# Network-based prediction of protein interactions

István A. Kovács[1,2,3], Katja Luck 🄳 [2,4], Kerstin Spirohn[2,4], Yang Wang[2,4], Carl Pollis 🄳 [2,4], Sadie Schlabach[2,4], Wenting Bian[2,4], Dae-Kyum Kim[2,5], Nishka Kishore[2,5], Tong Hao[2,4], Michael A. Calderwood 🄳 [2,4], Marc Vidal[2,4] & Albert-László Barabási[1,2,6,7]

Despite exceptional experimental efforts to map out the human interactome, the continued data incompleteness limits our ability to understand the molecular roots of human disease. Computational tools offer a promising alternative, helping identify biologically significant, yet unmapped protein-protein interactions (PPIs). While link prediction methods connect proteins on the basis of biological or network-based similarity, interacting proteins are not necessarily similar and similar proteins do not necessarily interact. Here, we offer structural and evolutionary evidence that proteins interact not if they are similar to each other, but if one of them is similar to the other's partners. This approach, that mathematically relies on network paths of length three (L3), significantly outperforms all existing link prediction methods. Given its high accuracy, we show that L3 can offer mechanistic insights into disease mechanisms and can complement future experimental efforts to complete the human interactome.

[1] Network Science Institute and Department of Physics, Northeastern University, Boston, MA 02115, USA. [2] Center for Cancer Systems Biology (CCSB), Dana-Farber Cancer Institute, Boston, MA 02115, USA. [3] Wigner Research Centre for Physics, Institute for Solid State Physics and Optics, H-1525 Budapest, P.O.Box 49 Hungary. [4] Department of Genetics, Blavatnik Institute, Harvard Medical School, Boston, MA 02115, USA. [5] Donnelly Centre, Toronto, Ontario, Canada, Department of Molecular Genetics, University of Toronto, Toronto, Ontario, Canada, Department of Computer Science, University of Toronto, Toronto, Ontario, Canada, Lunenfeld-Tanenbaum Research Institute, Sinai Health System, Toronto, ON, Canada. [6] Division of Network Medicine and Department of Medicine, Brigham and Women's Hospital, Harvard Medical School, Boston, MA, USA. [7] Department of Network and Data Science, Central European University, Budapest H-1051, Hungary. Correspondence and requests for materials should be addressed to I.A.K. (email: i.kovacs@northeastern.edu) or to A.-L.B. (email: a.barabasi@northeastern.edu)

As biological function emerges through interactions between a cell's molecular constituents, understanding cellular mechanisms requires a reasonably complete catalogue of all physical interactions between proteins[1–4]. Despite major efforts in high-throughput mapping[5–7], the number of missing human protein-protein interactions (PPIs) exceeds the experimentally documented interactions[8,9]. Consequently, computational tools are increasingly used to predict undetected, yet potentially biologically relevant interactions[10–12]. For proteins with well-described 3D structure, molecular dynamics simulations and machine learning techniques can predict de novo PPIs[10,11,13,14], but novel interactions are also inferred from co-expression profiles and other measures of functional similarity[15]. As a prominent example, PrePPI combines structural, sequence and additional biological evidences to predict PPIs at a genome-wide scale[14]. Yet, retest rates of high confidence PrePPI PPIs in experimental pairwise tests are several folds lower than the retest rates of curated interactions from the literature[5].

Alternatively, the increasing coverage of the interactome has inspired the development of network-based algorithms, which exploit the patterns characterizing already mapped interactions to identify missing interactions[16–18]. Such state-of-the-art network-based link prediction algorithms rely on the triadic closure principle (TCP)[10] (Supplementary Table 1), rooted in social network analysis, namely the observation that the more common friends two individuals have, the more likely that they know each other (neighborhood based similarity)[19–21]. Therefore, TCP-based algorithms assign a higher prediction score to protein pairs that share more of their interaction partners (Fig. 1a). As shown below, despite the plausibility of TCP for social networks, it fails to capture the structural and evolutionary forces that govern PPI networks. Our results in this paper suggest that the failure of TCP is not algorithmic, but fundamental: the hypothesis that protein pairs with similar interaction partners should interact fails for most protein pairs.

## Results

**The TCP Paradox.** To investigate the validity of the TCP hypothesis, we measured the relative number of shared interaction partners of proteins X and Y using the Jaccard similarity $J = |N_X \cap N_Y| / |N_X \cup N_Y|$, where $N_X$ and $N_Y$ are the interaction

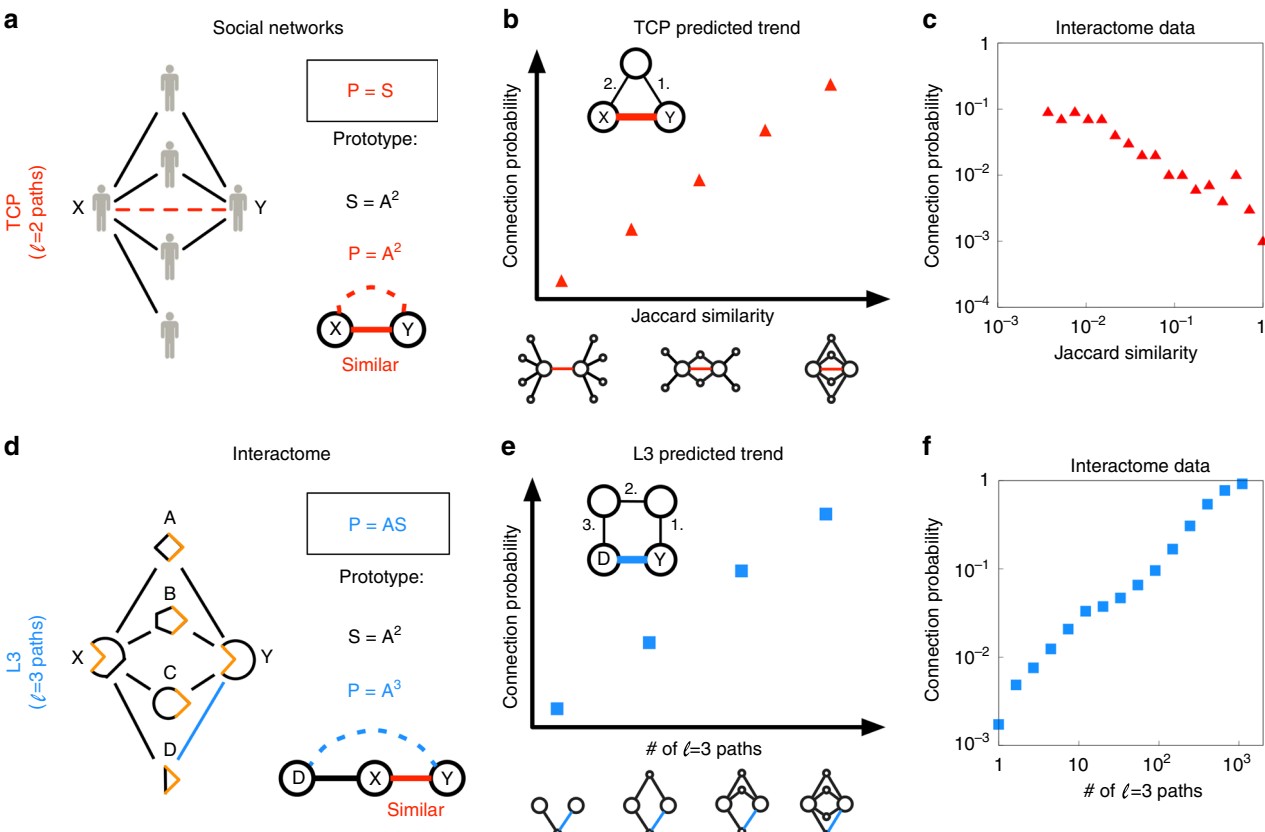

**Fig. 1** Network similarity does not imply connectivity. **a** In social networks, a large number of common friends implies a higher chance to become friends (red link between nodes X and Y), known as the Triadic Closure Principle (TCP). TCP predicts (P) links based on node similarity (S), quantifying the number of shared neighbors between each node pair ($A^2$). **b** A basic mathematical formulation of TCP implies that protein pairs of high Jaccard similarity are more likely to interact. **c** We do not observe the expected trend in Protein-Protein Interaction (PPI) datasets, as illustrated here for a binary human PPI network (HI-II-14)[5]: high Jaccard similarity indicates a lower chance for the proteins to interact (see Supplementary Fig. 3 for further networks). The data are binned logarithmically based on the Jaccard similarity values. **d** PPIs often require complementary interfaces[10,11], hence, two proteins, X and Y, with similar interfaces share many of their neighbors. Yet, a shared interface does not typically guarantee that X and Y directly interact with each other (see Supplementary Fig. 1 for an illustration with known 3D structures). Instead, an additional interaction partner of X (protein D) might be also shared with protein Y (blue link). Such a link can be predicted by using paths of length 3 (L3). L3 identifies similar nodes to the known partners (P = AS), going one step beyond the similarity-based argument of TCP. **e** Even without using any structural information, two proteins, such as Y and D are expected to interact if they are linked by multiple $\ell = 3$ paths in the network (L3). **f** As opposed to **c**, we observe a strong positive trend in HI-II-14 between the probability of two proteins interacting and the number of $\ell = 3$ paths between them, supporting the validity of the L3 principle

partners of X and Y. According to TCP, the higher the Jaccard similarity, the higher is the expected probability that X and Y interact (Fig. 1b). However, we find the opposite trend in all PPI networks, across several organisms (human, mouse, yeast, *C. elegans*, *D. melanogaster*, *A. thaliana*, *S. pombe*) and experimental mapping methods (binary interactions, pull-down, literature curation, see Supplementary Note 1 and Supplementary Table 2): The larger the Jaccard similarity between two proteins, the lower the chance that they interact with each other (Fig. 1c, Supplementary Fig. 3). In other words, the starting hypothesis of network-based PPI prediction tools cannot be validated. While the problem could lie with the limitations of existing network similarity measures[21], next we show that the failure of TCP is not rooted in the similarity measure we used, but it fails because it does not capture the biological principles that govern PPIs.

TCP reflects a long-standing tradition in link prediction, aiming to connect similar nodes[19,20]. Undeniably, interacting proteins do need to meet in the cell, both being present at least one cellular location and expressed together at least in one cellular state. Yet, beyond these basic factors, displaying similar co-localization and co-expression profiles is generally neither sufficient nor necessary. First, proteins can be highly similar in their sequence (e.g. certain paralogs) and still unable to interact, as they carry the same interfaces, instead of complementary ones (Fig. 1d). Second, highly similar proteins will likely share multiple additional partners. According to the TCP hypothesis, if proteins X and Y share multiple interaction partners (A, B, C), they likely interact with each other (Fig. 1a). From a structural perspective, the common partners simply reflect the fact that proteins X and Y have a similar or identical interaction interface, that recognizes the same binding sites in proteins A, B, and C (Fig. 1d). Yet, a common interaction interface of X and Y does not guarantee an interaction between X and Y. Instead, if protein X interacts with protein D, the network-based similarity of X and Y suggests that protein Y can also bind to protein D (Fig. 1d). We arrive to the same conclusion if we follow gene duplication events (Supplementary Fig. 2), that generate protein pairs V and V' that initially have an identical set of interaction partners, but do not imply that V and V' must interact with each other[22–24].

Here we propose that instead of finding similar candidates to a node, as done by TCP, to successfully predict PPIs, we must identify candidates that are similar to the known partners of a node. The simplest mathematical implementation of TCP relies on counting the shared neighbors of a pair of nodes, known as the Common Neighbors (CN) algorithm[25]. CN is quantified by $\mathcal{A}^2$, where $\mathcal{A}$ is the adjacency matrix (Fig. 1d), representing the hypothesis that proteins of multiple shared partners, i.e. those connected by paths of length two ($\ell = 2$), interact more frequently than unrelated proteins. Yet, the simplest implementation of the proposed paradigm is $\mathcal{A}^3$, utilizing paths of length three. Indeed, both structural and gene duplication arguments indicate that proteins linked by multiple paths of length $\ell = 3$, like proteins D and Y in Fig. 1e, are more likely to have a direct link (L3 principle). As a first test of the L3 principle, we measured the correlation between the number of $\ell = 3$ paths between a given protein pair, and the likelihood that they interact with each other (Fig. 1f). In contrast with Fig. 1c, that shows an anti-correlation between the paths of $\ell = 2$, documenting the failure of TCP for PPI networks, here we find a strict correlation (Fig. 1f), confirming the validity of the L3 principle. Next, we turn the L3 principle into a predictive tool, allowing us to experimentally test its predictive power.

**Degree-normalized L3 predictions**. High-degree nodes (hubs) induce multiple, unspecific shortcuts in the network, resulting in

biased predictions that can only be avoided by proper degree normalization. Such degree normalization is particularly important for L3, as it needs to choose candidates from nodes at $\ell = 3$ steps, an exponentially larger pool than the $\ell = 2$ distance pool utilized by TCP. To eliminate potential degree biases caused by intermediate hubs, we assign a degree-normalized L3 score to each node pair, X and Y

$$p_{XY} = \sum_{U,V} \frac{a_{XU} a_{UV} a_{VY}}{\sqrt{k_U k_V}}, \tag{1}$$

where $k_U$ is the degree of node U and $a_{XU} = 1$ if proteins X and U interact, and zero otherwise.

**Computational cross-validation**. To test the predictive power of L3, we need reliable network information. In the case of PPI networks, each data source comes with its limitations, prompting us to test L3 on multiple types of input networks. Literature curated interactomes of PPIs with multiple evidences have excellent replicability, but are impacted by selection biases[5]. We, therefore also consider interactomes emerging from systematic screens, that lack such biases[5–7]. Both approaches can be further split into binary or non-binary datasets, depending on the inclusion of co-complex membership annotations. We use these four classes of data, collected for humans, to compare L3 with the Common Neighbors algorithm[25], a common implementation of TCP (Fig. 2) that outperforms predictions based on the Jaccard similarity measure (Supplementary Fig. 5). For a Monte Carlo computational cross-validation we randomly split each network into a set of training PPIs and a test dataset, selecting 50% of the PPIs as the input network and measuring our ability to predict the remaining 50% (other fractions offer similar results and so does the leave-one-out cross-validation shown in Supplementary Fig. 10). Figure 2 shows the precision (fraction of validated protein pairs of the predictions) as a function of the recall (fraction of interactions covered from the test dataset), indicating that the predictive performance of L3 is about 2–3-times higher than TCP/CN for all datasets.

Starting from an input dataset of a binary interactome, the L3 algorithm allows us to predict more binary interactions (Fig. 2 and Supplementary Fig. 10A, C). On the other hand, starting from co-complex associations, L3 predicts further co-complex associations (Fig. 2 and Supplementary Fig. 10B, D). In principle, the input data can mix interactions from both sources, still leading to reliable predictions, without differentiating which category a specific prediction belongs to.

For completeness, we checked the performance of paths up to $\ell = 8$, finding that the best predictive power is indeed provided by $\ell = 3$ paths (Fig. 3a), supporting our structural and evolutionary arguments. Higher-order paths of odd length also do well, as they incorporate the strongly predictive $\ell = 3$ paths, taking additional steps back and forth along the same paths.

**High-throughput experimental validation**. To experimentally test the performance of L3, we started from the systematic binary interactome, HI-II-14[5]. We then tested the predicted interactions against a new systematic, binary, human PPI map, HI-III[26], resulting from an independent, high-throughput (HT) screen over the search space of ~18,000 × 18,000 human protein pairs. In Fig. 3b, we also compared L3 to the Preferential Attachment principle (PA)[25,27], a method not based on TCP. PA mimics unspecific binding between "sticky" proteins by placing a link between two nodes with a score given by the product of their degrees. PA is the simplest of the numerous alternative approaches to find a random benchmark with the observed node degrees as constraints[28]. Besides the computational cross-validation

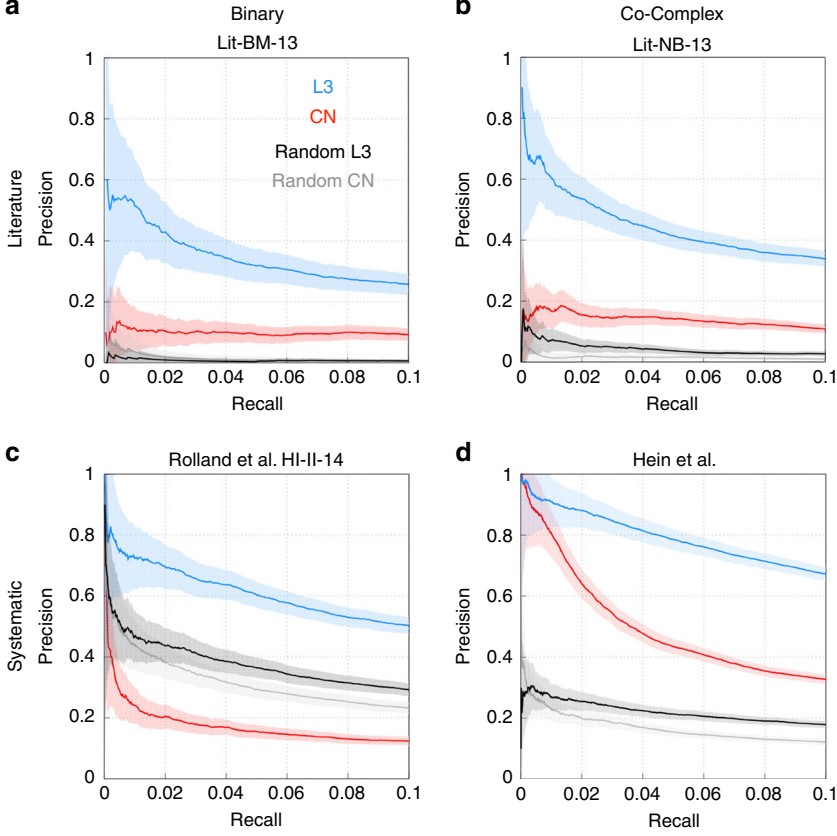

**Fig. 2** L3 outperforms Common Neighbors (CN) on PPI networks. Monte Carlo cross-validation of CN (a TCP implementation) and L3 on the four possible PPI data sources, arising from literature curation with multiple evidences (**a**, **b**[5]) or systematic screens (**c**[5], **d**[6]). We randomly select 50% of the PPIs and use it as the input network to predict the rest of the PPIs. Precision is the fraction of interacting proteins vs. all predicted pairs, while recall stands for the fraction of predicted PPIs compared to the number of test PPIs. We use all predictions until a 10% recall value is reached in each network. We find that L3 outperforms CN in all cases. We find qualitatively very similar results in a *k*-fold cross-validation scenario, as shown in the limit of an exhaustive leave-one-out cross-validation in Supplementary Fig. 10. In addition, we show the performance of both methods on randomized networks, where only the node degrees are preserved. L3 outperforms both these random benchmarks, irrespectively of the data source. In the case of the systematic binary network, HI-II-14, CN performs worse than in the randomized network, indicating a fundamental failure of TCP to capture the patterns shaping the underlying network structure. The shading around each curve indicates the standard deviation over 10 different random selections of the input PPIs. For additional datasets and validation see Supplementary Fig. 4

(Fig. 3b), we find that L3 also significantly outperforms CN and PA experimentally (Fig. 3c), along with 20 other published network-based link prediction methods (Supplementary Fig. 5). The substantial improvement of L3 holds for all tested organisms and data sources (Supplementary Figs. 4, 5 and Supplementary Table 3). Overall, the precision of L3 is seven fold higher than either CN or PA and more than twice the precision of the best performing literature method, Cannistraci Resource Allocation (CRA)[29] (Fig. 3c, d). CRA outperforms other TCP-based methods because it counts network motifs, that happen to contain both $\ell = 2$ and $\ell = 3$ paths (Fig. 3c), hence, it incorporates a subset of the L3 predictions. Yet, CRA's poorer performance compared to L3 is rooted in its reliance on $\ell = 2$ connectivity as well, that contributes false positives.

As negative control for the experiments, we randomly selected 100 non-interacting pairs, involving at least one of the proteins in the top 500 L3 predictions (RND). We did not experimentally recover any of these pairs in HI-III (Fig. 3d, Supplementary Note 1). Since HI-III is still incomplete, it is inherently unable to validate all protein pairs predicted by L3. We do want to know, however, what fraction of the L3 predicted pairs are potentially real. We therefore selected as positive control 100 interactions from HI-II-14, each involving at least one of the proteins in the top 500 L3 predictions (for details see Supplementary Note 1).

We find that 35% of these established interactions are recovered in the HT test. For the top 100 L3 predictions we find the same, 35% recovery rate in the HT test, indicating that the L3 predictions are recovered at the same rate as the already established PPIs. This can only happen if the vast majority (if not all) of the top L3 predicted interactions are true interactions, a conclusion supported by the pairwise testing described next.

Finally, to explore the robustness of L3, we tested if our results are tolerant against data incompleteness or noise (Fig. 3e, f). We find a stable precision up to removal of even 60–70% of the known interactions (false negatives, Fig. 3e). Furthermore, the predictive power of L3 persists even when the number of randomly added links exceeds the number of original links (false positives, Fig. 3f).

**Pairwise tests**. The HI-III-based experimental validation underestimates the performance of L3, as only a fraction of PPIs is found in a dataset from a HT screen[8]. To accurately assess the performance of L3, we also performed yeast-two-hybrid pairwise tests (PT) for the top 500 predicted links, utilizing—amongst others—the same positive and negative controls as above (Fig. 3g, Supplementary Note 1). Altogether, we performed ~3,000 pairwise tests, allowing us to classify each pair as either positive,

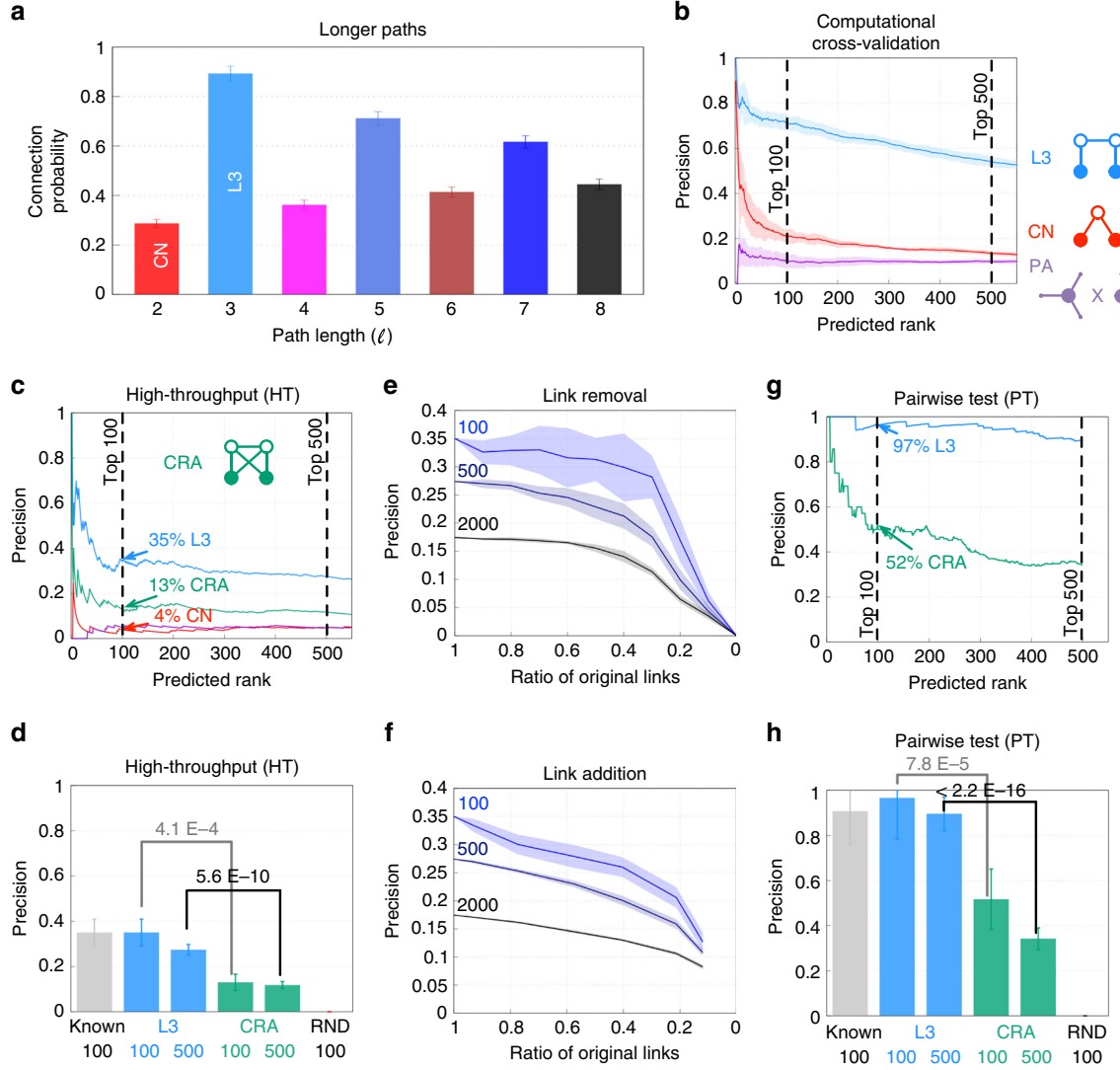

**Fig. 3** L3 is a precise and robust tool to find missing PPIs. **a** Connection probability in the top 1,000 HI-II-14[5] protein pairs ranked by different powers of the adjacency matrix, $\ell$, counting all paths of length $\ell = 2, \ldots, 8$. $\ell = 3$ paths are the most informative on direct connectivity. **b** In a 2-fold computational cross-validation on HI-tested (see the Methods section for details) L3 outperforms CN and PA at least three-fold. **c** In a high-throughput (HT) setting, we tested the L3 predictions on HI-tested, against the human interactome, HI-III[26]. L3 outperforms all other methods several fold, including the best performing literature method, CRA[29], out of 23 different methods tested (Supplementary Figures 5 and 6). **d** As a positive benchmark, we selected 100 known interactions (Known) and as a negative benchmark, 100 random pairs (RND), to set the expected window of precision values. For details see the Methods section. The recovery rate (precision) of L3 is significantly higher than that of CRA and comparable to the one of Known interactions (one-sided Fisher's exact test). **e** Robustness analyses of the L3 predictions with HT validation against data incompleteness, evaluated at the top 100, 500 and 2000 predictions, respectively. L3 is robust even when less then half of the PPIs are kept. **f** L3 is also robust against adding random links to the network, even when less then half of the links are PPIs. **g** Pairwise testing the top 500 predictions of L3 and CRA. We indicate the pairs where the experiments were conclusive (positive or negative) (Supplementary Note 1). **h** L3 not only outperforms CRA (one-sided Fisher's exact test), but the L3 predictions test positively with about the same rate as known interactions, indicating an optimal performance. Error bars indicate the expected standard deviation in **a**, **d** and **h**. The shading around each curve indicates the standard deviation over 10 realizations for **e** and **f**

negative or undetermined. We find that the recovery rate of 'Known' interactions—i.e. the fraction of positives over positives and negatives—has now increased to 91%, while none of the 'RND' control pairs scored positive (Fig. 3h). We also find a 90% recovery rate of the top 500 L3 predictions, which is once again 3-fold higher than that of CRA and it is comparable to the experimental recovery rate of the positive control (Fig. 3h).

To put these results into perspective, the primary high-throughput screens behind HI-II-14 resulted in ~35,500 PPI candidates, out of which ~14,700, or 41%, tested positively in the pairwise tests[5]. Current HT screens have an even higher validation rate, at least two-fold higher than the 34% predictive

precision of the best performing literature method, CRA (Fig. 3h). In contrast, 90% of the L3 predictions tested successfully by pairwise tests, in the same league as the best HT experimental pipelines. Yet, L3 requires high quality screening data as an input to achieve this remarkable performance. Taken together, the exceptional predictive power of L3 suggests that computational predictions have matured to the point that they can successfully augment HT screening in the search for new interactions.

Current quantitative link prediction methods are hard to put into practice, as it is up to the user to choose an acceptable compromise between precision and recall based on the ranked list provided. Here we go beyond this practice, as we not only provide

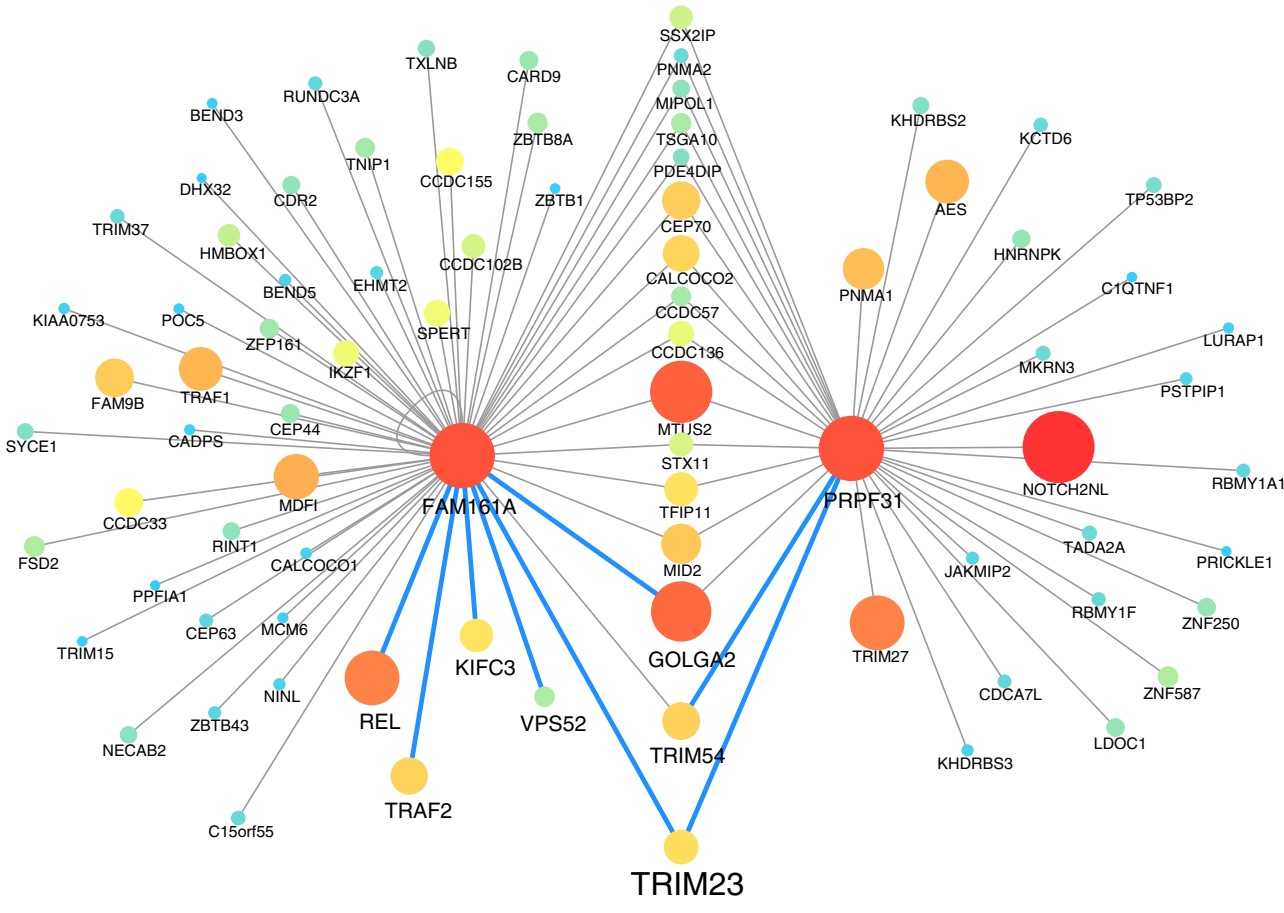

**Fig. 4** L3 provides mechanistic insights into protein function and complex diseases. For two proteins involved in retinitis pigmentosa (RP), FAM161A and PRPF31, we show all known interacting partners in HI-tested (gray), together with those predicted by the L3 algorithm and confirmed by pairwise tests (blue). The top L3 predicted interaction is connecting FAM161A to GOLGA2, two proteins without any shared interaction partners. The node size and color illustrates the degree of the proteins in HI-tested. In light of our experiments, GOLGA2, TRIM23, and TRIM54 are now amongst several shared interaction partners between FAM161A and PRPF31, a pre-mRNA splicing factor, whose mutations are causal for another form of RP[30]. This illustrates the key principle behind L3 (Fig. 1d), that two proteins, like FAM161A and PRPF31, despite sharing multiple interacting partners, do not necesseraly interact with each other, but share additional, previously unrecognized interaction partners

precision-recall curves, but also calculate the probability that each predicted interaction exists. This probability is obtained through a leave-one-out framework and we find that it is in excellent agreement with the experimental results (see the Methods section and Supplementary Fig. 8). Such reliable estimates help to select the number of predictions for downstream analyses and tests for the applications. For example, we estimate that starting from three existing screens of HI-III to predict new PPIs, at least another ~6,000 of the top 10,000 L3 predictions are true PPIs.

**Retinitis Pigmentosa.** Previously undetected PPIs can offer novel insights into disease mechanisms. Indeed, our top L3 prediction (Supplementary Fig. 9A) is an interaction between FAM161A and GOLGA2 (Fig. 4), which was found by HI-III and tested positively in our pairwise experiment as well. Family-based studies and homozygosity mapping have linked coding mutations in the gene FAM161A to a hereditary form of Retinitis Pigmentosa (RP), a retinal ciliopathy leading to progressive degeneration of photoreceptors. Yet, the cellular functions of FAM161A as well as

the molecular mechanisms leading to RP upon loss of FAM161A are largely unknown[30]. FAM161A localizes at the ciliary basal body and the connecting cilium of human photoreceptors[31,32], binds to microtubules[32] and has recently been found to interact with proteins of the Golgi apparatus (GA)[33]. The predicted and confirmed interaction between FAM161A and GOLGA2, a core member of the GA, offers a novel mechanistic insight for the role of FAM161A in GA function. Remarkably, FAM161A and GOLGA2 do not share any interaction partners, hence TCP-based algorithms are unlikely to predict this interaction. Within the top 500 L3 predictions we find five additional experimentally confirmed interaction partners of FAM161A: TRIM23, VPS52, KIFC3, TRAF2, and REL (Fig. 4), each of them offering insights into the function of FAM161A. TRIM23 and VPS52 both show GA localization and have been implicated in regulating vesicle trafficking between the GA and other cellular compartments. KIFC3, a recently identified interactor of FAM161A[33], is well known for its function as a microtubule motor protein. Our findings can also indicate novel directions, such as a potential link of FAM161A to NF-κB signaling. Indeed, TRIM23, TRAF2 and

REL have demonstrated roles in NF-$\kappa$B activation, which is particularly interesting in light of the activation of cell death pathways in RP-affected photoreceptors. Altogether, these newly identified interactions position FAM161A within a molecular network that connects GA function with cilium organization and intracellular transport, providing detailed insights into potential molecular mechanisms that underlie RP.

## Discussion

The exceptional success of the L3 framework is rooted in its ability to capture the structural and evolutionary principles that drive PPIs. Yet, it is also rooted in the fact that the already detected interactions, while offering an incomplete coverage of the full interactome, have reached a critical coverage and accuracy[34] to make future accurate predictions possible. Yet, in contrary of the current network paradigm, interacting proteins are not necessarily similar and similar proteins do not necessarily interact, questioning the traditional validation strategy based on biological similarity of the predicted protein pairs (Supplementary Fig. 7). Our systematic high-throughput validations show that the L3 principle significantly outperforms all existing link prediction methods, as well as state-of-the-art bioinformatics tools, PrePPI[14] and STRING[35], that rely on additional biological information. For example, we calculate the validation rate of PrePPI by checking the fraction of positives in the same prediction space of protein pairs. The top 500 PrePPI predictions test positively in only 2.8% of the cases. We also evaluated protein interactions from the STRING database, leading to a recovery rate of 1.8% for the top 500 pairs. On the contrary, our results suggest that most (if not all) of the top L3 predictions are true PPIs, amenable to further detailed studies, such as interface prediction[36].

Despite its exceptional predictive power, the L3 framework is not without limitations. First, like all network-based methods, it cannot find interacting partners for proteins without known links. For such proteins, we can, however, seamlessly, integrate into the L3 framework information on sequence, evolutionary history or 3D structure, used by some PPI prediction algorithms[10,11,13,14,37]. Second, L3 is most probably only the first step in our journey to further improve the performance of PPI predictions. Deriving better degree normalizations, or combining $\ell = 3$ with additional information provided by longer paths (Fig. 3a) can potentially offer further improvements[38,39]. Finally, the high retest rate of L3 predictions indicates that network-based predictive algorithms are poised to complement future mapping efforts. Taken together, L3, coupled with experimental validation, offers a powerful and necessary tool for the completion of the human interactome, allowing us to exploit network effects as we aim to uncover the mechanistic roots of human disease[7,34–38].

## Methods

**Pairwise testing experiments**. We tested experimentally the top 500 predictions of L3 against the top 500 predictions of CRA[29] on the human network, HI-tested. HI-tested is a subset of the human interaction dataset HI-II-14[5], restricted to a single ORF for each gene, present in HI-III[26], and leaving out keratins (KRT*). To test the overall efficiency of the experiment, we included 94 literature curated interactions with multiple evidence (Lit-BM-13[5]), as well as a set of 88 positive reference interactions from the literature (PRS). In principle, the proteins of our top predictions might have special characteristics, which locally modify the recovery rate of their interactions. To have a more specific assessment, our selected positive set ("Known" PPIs) contains 100 randomly selected known links in HI-tested, connected to at least one of the nodes in the top 500 L3 predictions. To control the false positive rate, we selected a set of 144 random node pairs in the random reference set (RRS) and a more specific set of 100 random pairs involving the proteins in the top 500 L3 predictions (RND). Altogether, we pairwise tested 1,485 non-redundant pairs, in two orientations, classifying each pair as either positive, negative or undetermined. Details of the experimental protocol are summarized in Supplementary Note 1.

**Statistical analysis**. All statistical analyses were performed using the R package (v3.2.3, http://www.r-project.org/). Details of the quantitative performance evaluation are listed in Supplementary Note 2.

**Assigning probabilities to the L3 predictions**. To estimate the probability of each predicted link to test positively in a pairwise testing experiment, we first calculated a prediction score for each existing link in HI-tested in a leave-one-out scenario and ranked these known PPIs together with the newly predicted links based on their prediction scores. Assuming that around a given rank pairs are more likely to be real if surrounded by already known links, we estimate the validation probability at a given rank by calculating the ratio of known interactions to all pairs in a window of ±50 known interactions around the studied rank. We then sum up these probabilities until a given rank, providing the expected number of validated interactions up to that rank.

**Code availability**. The L3 prediction code, together with example datasets, input data files and predictions, is available at [https://doi.org/10.5281/zenodo.2008592]. Further codes written for and used in this study are available from the corresponding author upon reasonable request.

**Reporting Summary**. Further information on experimental design is available in the Nature Research Reporting Summary linked to this article.

## Data availability

Current version of the human interactome, HI-III, is available at Ref. [26]. The version used in this study (screens 1-3) and additional datasets are available at [https://doi.org/10.5281/zenodo.2008592] and at the original references. The protein interactions from this publication have been submitted to the IMEx (http://www.imexconsortium.org) consortium through IntAct[44] and assigned the identifier IM-26274 [https://www.ebi.ac.uk/intact/search/do/search?searchString=pubid:IM-26274].

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

## Acknowledgements

We thank D. Hill, F.P. Roth, F. Cheng and M. Santolini for useful discussions on the manuscript and A. Grishchenko for help with visualization. This work was supported by an NHGRI Center of Excellence in Genome Science grant P50HG004233 to M.V., and A.-L.B; an NHGRI U41HG001715 awarded to M.V., and M.A.C.; and an NIGMS R01GM109199 awarded to M.A.C. A.-L.B. and I.A.K were also funded by P01HL132825. We gratefully acknowledge the support of The National Human Genome Research Institute (NHGRI) and National Institute of General Medical Sciences (NIGMS) of NIH.

## Author contributions

Computational analyses were developed and performed by I.A.K. Experiments were designed by M.A.C., K.S., K.L. and I.A.K and performed by K.S., C.P., S.S. D.-K.K., N.K. performed sequence confirmation. W.B. processed Sanger sequencing data and T.H. designed the cherrypicking files. Y. W. carried out the protein structural analyses. Principal investigators overseeing primary data management, structural biology and experiment evaluation were A.-L.B., M.V. and M.A.C. A.-L.B. directed the overall research effort. I.A.K., K.L. and A.-L.B. wrote the manuscript with contributions from other co-authors.

## Additional information

**Competing interests:** A.-L.B. is a co-founder of Scipher Medicine, a startup that uses network concepts to explore human disease. The remaining authors declare no competing interests.

