## [Peer Review File · Nature Communications]

Reviewers' comments:

Reviewer #1 (Remarks to the Author):

The manuscript 'Network-based prediction of protein interactions' by Kovacs et al. deals with a novel network topology based method allowing the authors to predict potential protein-protein interactions. The authors used the (incomplete) humans interactome and showed that their approach outperforms other, previous approaches. At the heart of the approach is the observation that the TCP (triadic closure principle known from social network analysis that indicates possible social ties) does not apply to the presence of potential protein interactions. Such proteins would have common interacting neighbors, suggesting that such protein pairs are rather sharing similar folds that exclude their ability to interact directly. In turn, the authors suggest that proteins 3 steps away from a protein in question are potential interactors. In a series of computational analyses of existing large-scale human interactions they show the efficacy of their new approach. In addition, the authors also apply the approach on a previously unpublished, comprehensive human interaction data set that corroborates the relevance of their approach. Furthermore, the authors compare their results with other methods and measures, convincingly showing that their concept allows the prediction of potentially interacting protein pairs.

The set-up of this study and the results are very convincing. However, there are still a couple of points that the authors should consider:

- As it reads right now the authors calculate a L3 score for each pair of non-interacting pairs that are 3 steps away from each other in the underlying network. In other words, the score normalizes the contribution of each path of length 3 to the underlying score. Scores thus obtained are ranked and the top ranked pairs were investigated toward their ability to indeed predict a potential interaction. As a consequence it is up to the user to decide which threshold points to interactions. As this threshold is arbitrary, it would be good for the authors to discuss what they consider a good cut-off to obtain interaction candidates.

Minor points:

- In Fig. 2, it is a bit unclear what predictions were used to find the precision-recall curves. Where predictions up to a given threshold of the score used or did the authors use bins of certain scores to find these curves?
- In Fig. 3C it is unclear what the different curves relate too. In the legend it sounds like that these curves refer to the results of the CRA method and that it outperforms all other methods including L3. In the main text, however, L3 is mentioned as the best performing method (better than CRA). So, clarification is needed.
- Also, results in Fig. 3D are confusing. It is clear that L3 can recover 35% of a random sample of known human protein interactions if one interacting protein is involved in the prediction. However, what is the meaning and relevance of precision of the known interactions in Fig. 3D and H? Furthermore, Fig. 3D is mentioned after Figs. 3E-F, hampering the flow of the paper a bit.

Reviewer #2 (Remarks to the Author):

The manuscript presents an interesting network-based method for predicting new protein-protein interactions. The method is elegant and makes biological sense, and its efficacy has been validated both computationally and experimentally. The manuscript is well written and easy to follow. The description of the methods and data sources is very clear. All calculations and use of statistics throughout the manuscript were properly carried out.

I only have some minor questions and comments:

1. In Fig. 1C and Fig. S3, the authors show that the Jaccard similarity between two proteins correlates negatively with the probability that they interact with each other. While the overall trend holds for all interactomes examined, there is a consistent peak at around Jaccard index = 0.4 across all interactomes. Is there a reason behind that?

2. The authors claim to have used cross-validation to computationally evaluate the performance of their L3 method. However, what the authors did was randomly splitting the data 10 times, each time using 50% of the data for training and the other 50% for testing, which is technically not cross-validation. In a k-fold cross-validation setting, the data is evenly split into k folds, and each time 1 of the folds is used for testing while the other k-1 folds are used for training. In this way each instance is tested once. Would the trends in Fig. 2 change if say, 5-fold cross validation is used?

3. To put the method proposed into use, the authors should provide a list of newly predicted protein-protein interactions based on a comprehensive input interactome.

4. I am wondering whether these predicted interactions are binary physical interactions or more likely to be co-complex associations (both contain useful information). Based on their validation experiments, the answer seems to be the former. If so, would it be reasonable to apply interface prediction algorithms (e.g., Meyer et al, Nature Methods, 2018, 15:107-114) to these predicted interactions. Can authors comments on these points?

5. Minor formatting issue: In Fig. S4, the color of A3 does not match between the text and the plots.

Manuscript #: NCOMMS-18-29503

Response to Reviewer #1:

The manuscript Network-based prediction of protein interactions by Kovacs et al. deals with a novel network topology based method allowing the authors to predict potential protein-protein interactions. The authors used the (incomplete) humans interactome and showed that their approach outperforms other, previous approaches. At the heart of the approach is the observation that the TCP (triadic closure principle known from social network analysis that indicates possible social ties) does not apply to the presence of potential protein interactions. Such proteins would have common interacting neighbors, suggesting that such protein pairs are rather sharing similar folds that exclude their ability to interact directly. In turn, the authors suggest that proteins 3 steps away from a protein in question are potential interactors. In a series of computational analyses of existing large-scale human interactions they show the efficacy of their new approach. In addition, the authors also apply the approach on a previously unpublished, comprehensive human interaction data set that corroborates the relevance of their approach. Furthermore, the authors compare their results with other methods and measures, convincingly showing that their concept allows the prediction of potentially interacting protein pairs. The set-up of this study and the results are very convincing.

We wish to thank the Reviewer for the excellent summary of our results and for the careful reading of the manuscript.

However, there are still a couple of points that the authors should consider:

1. As it reads right now the authors calculate a L3 score for each pair of non-interacting pairs that are 3 steps away from each other in the underlying network. In other words, the score normalizes the contribution of each path of length 3 to the underlying score. Scores thus obtained are ranked and the top ranked pairs were investigated toward their ability to indeed predict a potential interaction. As a consequence it is up to the user to decide which threshold points to interactions. As this threshold is arbitrary, it would be good for the authors to discuss what they consider a good cut-off to obtain interaction candidates.

The Reviewer raises an important point that all quantitative link prediction methods struggle with, namely that it is up to the user to choose an acceptable compromise between precision and recall based on the provided ranked list. Here we went beyond this practice, as we not only provide precision-recall curves, but also calculate the probability that each predicted interaction exists. This probability is obtained through a leave-one-out framework and we showed that it is in excellent agreement with the experimental results, as discussed on page 5 and Fig. S8. We now added more details on this important point to page 5. These probabilities allow us to make a much more informed decision for the

experimental threshold based on resource limitations. We even made a specific suggestion on this threshold, to test the top 10,000 L3 predictions on HI-III (page 5), of which about 6,000 are expected to be true PPIs.

Minor points:

2. In Fig. 2, it is a bit unclear what predictions were used to find the precision-recall curves. Were predictions up to a given threshold of the score used or did the authors use bins of certain scores to find these curves?

We use all predictions, ranked by the L3 prediction score, until the shown recall value of 0.1. There is no additional threshold or binning applied. In the revised version we made this explicit in the figure caption.

3. In Fig. 3C it is unclear what the different curves relate to. In the legend it sounds like that these curves refer to the results of the CRA method and that it outperforms all other methods including L3. In the main text, however, L3 is mentioned as the best performing method (better than CRA). So, clarification is needed.

We thank the Reviewer for this observation. In all figures, results for L3 are shown in blue, which we tried to indicate in Fig. 3 in panels A, B D & H. In Fig. 3C, D & H, results for CRA are shown in green. Although CRA outperforms all other tested methods from the literature, L3 outperforms CRA, delivering 2-3 fold higher precision. In the revised version, we improved the clarity of the figure labels and the legend, and we wish to thank the Reviewer for bringing the lack of clarity in this figure to our attention.

4. Also, results in Fig. 3D are confusing. It is clear that L3 can recover 35% of a random sample of known human protein interactions if one interacting protein is involved in the prediction. However, what is the meaning and relevance of precision of the known interactions in Fig. 3D and H? Furthermore, Fig. 3D is mentioned after Figs. 3E-F, hampering the flow of the paper a bit.

We do realize that the experimental test, correctly done with all necessary controls, can appear sometimes confusing, so we appreciate the Reviewer's effort to bring our attention to areas where we lack sufficient clarity. In Fig. 3D we include the same main results as in Fig. 3C for the top 100 and 500 predictions, respectively. The precision shown in the panel is the recovery rate of various sets of protein pairs in the high-throughput (HT) experimental data. The reason to also include 'Known' PPIs is to set a positive control for the experiment, measuring the precision value of 100% true PPIs, as discussed in page 5 of the 'High-Throughput Experimental Validation' section. In Fig. 3D, 'Known' interactions from HI-II are recovered with a 35% rate, quantifying the best precision expected in this HT experiment. We find that the top L3 predictions are recovered with the same rate, indicating that in principle the L3 predictions are just as accurate as

gold standard, known PPIs. In other words, remarkably, all top L3 predictions are expected to be valid. Without the positive benchmark of 'Known' interactions we could not assess the true accuracy of L3. Similarly, in Fig. 3H, 'Known' interactions are recovered with an 80% rate, setting the maximum performance in the pairwise testing experiment (see page 5). Once again, top L3 predictions have the same, optimal precision. We appreciate the Reviewer's comment, so we improved the clarity of the figure legend. As in the submitted version, the figure panels are mentioned in order, Fig. 3D before Figs. 3E-F, in the bottom of page 4.

In summary, we would like to thank the Reviewer's careful reading of the manuscript and the constructive comments that helped us to significantly improve the presentation of our results, including polishing the presentation of the text, highlighted in blue.

Response to Reviewer #2:

The manuscript presents an interesting network-based method for predicting new protein-protein interactions. The method is elegant and makes biological sense, and its efficacy has been validated both computationally and experimentally. The manuscript is well written and easy to follow. The description of the methods and data sources is very clear. All calculations and use of statistics throughout the manuscript were properly carried out.

We thank the Reviewer for the appreciation and the positive feedback on our work, especially for his/her appreciation of the experimental part.

I only have some minor questions and comments:

1. In Fig. 1C and Fig. S3, the authors show that the Jaccard similarity between two proteins correlates negatively with the probability that they interact with each other. While the overall trend holds for all interactomes examined, there is a consistent peak at around Jaccard index = 0.4 across all interactomes. Is there a reason behind that?

This is a good point that in our view lies in the limitations of the Jaccard index. As we now discuss in page 2, lacking of a perfect similarity measure, we use the Jaccard similarity, which however, is not without limitations. It's major limitation lies in it's difficulty to quantify similarity between low-degree proteins. Indeed, in the figures highlighted by the Reviewer, we observe a peak at the bin around the value 0.5. This value is special in the sense that it can happen for any protein degree higher than 1. Other values of the Jaccard index cover far less protein pairs. For example, all values between 0.5 and 1 are impossible for degree 1 and degree 2 proteins. The fat tailed nature of the degree distribution of PPI networks means an abundance of low-degree proteins, further enhancing the concentration of protein pairs on Jaccard index=0.5. Altogether, we observe a concentration on Jaccard index=0.5 of not only PPI pairs, but also the signal from many of the triangles that they are involved in. Although measuring node similarity adequately and assessing the extent of degree-sensitivity of various similarity measures is an important open problem, it does not affect our main findings.

2. The authors claim to have used cross-validation to computationally evaluate the performance of their L3 method. However, what the authors did was randomly splitting the data 10 times, each time using 50% of the data for training and the other 50% for testing, which is technically not cross-validation. In a k-fold cross-validation setting, the data is evenly split into k folds, and each time 1 of the folds is used for testing while the other k-1 folds are used for training. In this way each instance is tested once. Would the trends in Fig. 2 change if say, 5-fold

cross validation is used?

We appreciate the Reviewer's attention to this important detail of our work. Indeed, as stated in Fig. 2, for simplicity and flexibility, we did not perform a 2-fold cross-validation, but we utilized a Monte Carlo cross-validation, where each split is used only once. Apart from its simplicity, a major advantage of a Monte Carlo framework is that the complete freedom in choosing the size of the training data, a point that becomes important when dealing with highly incomplete data.

We did perform, however, a true 2-fold cross-validation, whose results are shown in Fig. 3B. We have now clarified this specific point in the revised manuscript. Generally, we find that our results are qualitatively unchanged for k-fold cross-validation. To illustrate this, we added a new figure to the SI, Fig. S10, indicating the results for the limiting case of an exhaustive leave-one-out cross-validation, with all links tested one-by-one. The results are qualitatively the same as those in Fig. 2, although the top precision is bounded by data incompleteness.

3. To put the method proposed into use, the authors should provide a list of newly predicted protein-protein interactions based on a comprehensive input interactome.

We thank the Reviewer for this great suggestion. The L3 method is relatively easy to implement and quick to run on a desktop computer for any dataset, and we fully agree that it would help its diffusion if we made both our code and our predictions easily accessible. Our input networks, the L3 prediction code and a library of predictions will be shared through Github under the following DOI: [10.5281/zenodo.2008592](https://doi.org/10.5281/zenodo.2008592)

In addition, we have uploaded our predicted and validated PPIs to IntAct, available under the new Ref. [45]. We have extended the Code and Data availability sections of the manuscript accordingly, and wish to thank the Reviewer for prompting us to do this.

4. I am wondering whether these predicted interactions are binary physical interactions or more likely to be co-complex associations (both contain useful information). Based on their validation experiments, the answer seems to be the former. If so, would it be reasonable to apply interface prediction algorithms (e.g., Meyer et al, Nature Methods, 2018, 15:107-114) to these predicted interactions. Can authors comments on these points?

This is a very good and important point. Our results suggest (see Fig. 2 and Figs. S4 & S10) that L3 can predict both binary and co-complex interactions. The nature of the predictions is expected to match that of the input data. Starting from a network of binary interactions, we predict more binary interactions (Figs. 2 and S10, panels A&C). On the other hand, starting from co-complex associations, we are able to predict further co-complex associations (Figs. 2 and S10, panels B&D). If the input data were a mixture of the two, for example, a literature curated integrated dataset, then the predictions would also be expected to be mixtures,

without an indication of which category a specific prediction belongs to. In the revised manuscript, we have now extended the Computational Cross-validation section accordingly. In this work, our experimental efforts focused on binary physical interactions. Indeed, for these binary predictions, it makes perfect sense to apply interface prediction methods, to gain further insights on the interactions. In the revised manuscript we extended the Discussion section with this future direction, and we also added a reference to the Meyer et al. paper.

5. Minor formatting issue: In Fig. S4, the color of A3 does not match between the text and the plots.

We double checked our figures and are not seeing a mismatch between the mentioned and shown color for the unnormalized A3 score (brown) in Fig. S4. However, we noticed a typo in Fig. S7 and fixed it in the revised version.

Finally, we would like to thank the Reviewer for the constructive, detailed comments helping to increase the clarity of our results, and their accessibility to the community.

REVIEWERS' COMMENTS:

Reviewer #1 (Remarks to the Author):

The manuscript ' Network based prediction of protein interactions' by Kovacs et al. has been revised. So far the authors answered the questions and concerns of this reviewer. However, a couple of really minor issues (that the authors can be trusted to sort out) remain:

- The prediction of potential interactions is done by randomly splitting known interactions in two halves, where one half is being used to predict interactions as a 'training' set (using L3 and other methods for comparison) while the second half is used as a 'validation' set allowing the authors to find precision and recall values (such as in Figures 2 and 3A). In Figure 3B the legend says that it was done with the 2-fold cross-validation method (as described above). However, the main text indicates that HI-II was used for the prediction and HI-III data was used for validation. It seems that the HI-II data was used for the prediction and HI-III data was used for validation in Fig. 3C. I hope I am not misconstruing something here, so I'd urge the authors to check if there is no problem there.

- In Fig. 3D, the authors use as a positive benchmark 100 known interactions and check if the HT experimental screen (HI-III) picked them up. The same was done with randomly sampled interactions. Oddly, the paragraph that deals with this is found after the results in Fig. 3CD were presented. Also, such interactions were used to benchmark pairwise testing in Fig. 3H. So, some rearranging this paragraph would help the flow of the paper.

- As for a stylistic point (that is up to the editor), the shaded boxes around Figs. 3CD, 3EF and 3GH more look like that they were used for slides at a talk and may not fit a journal figure.

Other than that the manuscript is in great shape and good to go!

Reviewer #2 (Remarks to the Author):

The authors have addressed my comments, I have no more concerns.

Manuscript #: NCOMMS-18-29503B

Response to Reviewer #1:

The manuscript ' Network based prediction of protein interactions' by Kovacs et al. has been revised. So far the authors answered the questions and concerns of this reviewer. However, a couple of really minor issues (that the authors can be trusted to sort out) remain:

1. The prediction of potential interactions is done by randomly splitting known interactions in two halves, where one half is being used to predict interactions as a 'training' set (using L3 and other methods for comparison) while the second half is used as a 'validation' set allowing the authors to find precision and recall values (such as in Figures 2 and 3A). In Figure 3B the legend says that it was done with the 2-fold cross-validation method (as described above). However, the main text indicates that HI-II was used for the prediction and HI-III data was used for validation. It seems that the HI-II data was used for the prediction and HI-III data was used for validation in Fig. 3C. I hope I am not misconstruing something here, so I'd urge the authors to check if there is no problem there.

As the Reviewer correctly stated, in Figure 3B we use a 2-fold cross-validation, while in Fig. 3C we present the results of the High-Throughput validation, using the experimental dataset, HI-III, for validation. We have made the presentation more clear.

2. In Fig. 3D, the authors use as a positive benchmark 100 known interactions and check if the HT experimental screen (HI-III) picked them up. The same was done with randomly sampled interactions. Oddly, the paragraph that deals with this is found after the results in Fig. 3CD were presented. Also, such interactions were used to benchmark pairwise testing in Fig. 3H. So, some rearranging this paragraph would help the flow of the paper.

We thank the Reviewer for this comment that helped us to improve the clarity of our presentation by rearranging our results.

3. As for a stylistic point (that is up to the editor), the shaded boxes around Figs. 3CD, 3EF and 3GH more look like that they were used for slides at a talk and may not fit a journal figure.

We have updated the figures and removed the shaded boxes. We have also updated panels G and H in Fig. 3 by applying a more stringent filtering on the negative pairs, leading to slightly improved retest rates, within the previous error bars. All our previous conclusions hold and remain statistically significant.

Other than that the manuscript is in great shape and good to go!

We wish to thank the Reviewer again for the careful reading of the manuscript.